# Identification and Characterization of *LaSCL6* Alleles in *Larix kaempferi* (Lamb.) Carr. Based on Analysis of Simple Sequence Repeats and Allelic Expression

**Qiao-Lu Zang, Xiang-Yi Li, Li-Wang Qi and Wan-Feng Li \***

State Key Laboratory of Tree Genetics and Breeding, Key Laboratory of Tree Breeding and Cultivation, National Forestry and Grassland Administration, Research Institute of Forestry, Chinese Academy of Forestry, Beijing 100091, China; zangql@caf.ac.cn (Q.-L.Z.); lixy.caf@foxmail.com (X.-Y.L.); lwqi@caf.ac.cn (L.-W.Q.)
\* Correspondence: liwf@caf.ac.cn; Tel.: +86-10-62889628

**Abstract:** Simple sequence repeats (SSRs) are widely used as markers for the assessment of genetic diversity and marker-assisted breeding. In a previous study, two SSRs (GCA and CCA), were found in the genomic sequence of *Larix* (*La*) *SCL6*, which plays important roles in the growth and development of *Larix kaempferi* (Lamb.) Carr. In this study, we analyzed the polymorphisms of these two SSRs in the *L. kaempferi* population. We found that each SSR had five different polymorphisms, among which $(GCA)_7$ and $(CCA)_7$ were predominant. In addition, 12 haplotypes were detected, with $(GCA)_7(CCA)_7$ having the highest frequency. Furthermore, we detected the haplotypes of *LaSCL6* in mature trees and their seeds and analyzed the relationships between parents and offspring. The expression patterns of five *LaSCL6* alleles were analyzed and they showed balanced expression during vegetative development. Taken together, these findings not only provide more genetic information on *LaSCL6*, but also provide a candidate marker for genetic studies and breeding.

**Keywords:** Japanese larch; endosperm; haplotypes; genetics; breeding

## 1. Introduction

Simple sequence repeats (SSRs) are a class of DNA sequences consisting of short, tandem-repeated motifs (1–6 bp in length) [1]. SSR sequences are highly polymorphic, and their presence results in allelic variants with different frequencies at the population level [2–4]. In addition, some allelic variants show differential expression during plant growth and development [5–7]. Due to their advantages of being co-dominant, multi-allelic, reliable, PCR-based, and abundant in plant genomes [8], SSRs have been used as markers for the assessment of genetic diversity [9,10], genetic structure [11], parentage analyses [12], pedigree and mating system analyses [13], and marker-assisted breeding [14].

*Larix kaempferi* (Lamb.) Carr, a forest tree of important ecological and economic value and widely-grown in the northern hemisphere, is monoecious and mainly wind-pollinated [15]. In recent years, SSRs have been used to investigate the pollen contamination rate and paternal contributions of *L. kaempferi*, which affects the seed productivity and severely limits the improvement of seed orchard yield and the construction of clonal seed orchards [16,17].

*SCARECROW-LIKE6* (*SCL6*), a member of the *GRAS* (*GAI-RGA-SCR*) family that is regulated by microRNA171 at the post-transcriptional level, is involved in many aspects of growth and development, such as shoot branching [18], meristem maintenance [19], and somatic embryogenesis [20,21]. In a previous study, we identified the homologue of *SCL6* in *L. kaempferi*, and we found two SSRs (GCA and CCA) in the genomic sequence of *LaSCL6* [22]. Understanding the genetic variation of a specific gene in a population is important for the study of its functions and for future breeding improvements.

In this study, we analyzed the polymorphism of SSR sequences of *LaSCL6* in the *L. kaempferi* population, and we studied the specific expression of five alleles. Our results not only provide more genetic information on *LaSCL6*, but also provide a candidate marker for genetic studies and breeding.

## 2. Materials and Methods

### 2.1. Plant Materials

All the materials were collected from a Dagujia seed orchard (42°22′ N, 124°51′ E), Liaoning Province, in Northeast China. Endosperms were used to assess the frequency of SSRs and they were separated from 200 mature seeds; after immersion in water for one day, the seed coats were removed, and then the endosperms were isolated for DNA extraction. The needles and seeds of three mature trees were used to study the genetic information delivery from parents to offspring, and endosperms and embryos were sampled separately; the needles, endosperms, and embryos were used for DNA extraction. Six-month-old seedlings were used to study the expression of alleles. After analyzing the alleles of 40 seedlings, the needles, stem, and the root from a single seedling were sampled for RNA extraction. All the samples used for DNA and RNA extraction were frozen in liquid nitrogen and stored at −80 °C.

### 2.2. DNA Extraction and Polymerase Chain Reaction (PCR) Amplification

The genomic DNA of *L. kaempferi* was isolated with the CTAB plant genome DNA rapid extraction kit (Aidlab Biotech, Beijing, China) according to the manufacturer's protocols. The primers 5′-AGCGAGGTCAAGAAAGAAGAGC-3′ and 5′-TTGGGAACGAATGGCGTAGGG-3′ were used to amplify the sequence fragment containing two SSR loci from the DNA template with Platinum® Taq DNA polymerase (Invitrogen, Carlsbad, CA, USA). The PCR products were purified with a gel extraction kit (Tiangen, Beijing, China), ligated into the pEASY®-T1 simple cloning vector (TransGen Biotech, Beijing, China), and sequenced. Multiple sequence alignments were performed with ClustalX [23]. The tertiary structures were predicted by SWISS-MODEL (https://swissmodel.expasy.org/).

### 2.3. RNA Extraction and Quantitative Reverse Transcription Polymerase Chain Reaction (qRT-PCR)

Total RNAs extracted from the needle, stem, and root were isolated with the EasyPure RNA kit (TransGen Biotech, Beijing, China) according to the improved manufacturer's protocol and then reverse-transcribed into cDNA with the TransScript® II one-step gDNA removal and cDNA Synthesis SuperMix kit (TransGen Biotech, Beijing, China). TB Green® Premix Ex Taq™ (Tli RNase H Plus) (Takara, Shiga, Japan) was used to assess the expression levels of alleles with the allele-specific primers. *LaEF1A1* was used as the internal control [21]. The qRT-PCR was performed with three biological replicates and the data are shown as mean ± SD. Statistical analysis was performed with SPSS19.0 using analysis of variance.

## 3. Results and Discussion

### 3.1. SSRs in LaSCL6 Show High Levels of Polymorphism

In 200 endosperms, four types of GCA repeats and three types of CCA repeats were detected, and the frequencies of $(GCA)_7$ (161/200, 80.5%) and $(CCA)_7$ (197/200, 98.5%) were the highest (Table 1). In the 40 seedlings, five types of GCA repeats and five types of CCA repeats were detected, and the repeats $(GCA)_7$ (51/80, 63.8%) and $(CCA)_7$ (62/80, 77.5%) were also most frequent (Table 1).

**Table 1.** Polymorphism of simple sequence repeats in *LaSCL6* in the population of *Larix kaempferi* (Lamb.) Carr.

| Repeat Motif | Number of Repeats | Frequency in Endosperms | Frequency in Seedlings |
|---|---|---|---|
| GCA | 4 | 5/200 (2.5%) | 6/80 (7.5%) |
| | 6 | 2/200 (1.0%) + 15/200 (7.5%) [a] | 13/80 (16.3%) + 4/80 (5.0%) [a] |
| | 7 | 161/200 (80.5%) | 48/80 (60.0%) + 3/80 (3.8%) [b] |
| | 8 | – | 1/80 (1.3%) |
| | 9 | 17/200 (8.5%) | 5/80 (6.3%) |
| CCA | 6 | – | 1/80 (1.3%) |
| | 7 | 197/200 (98.5%) | 62/80 (77.5%) |
| | 8 | 2/200 (1.0%) | 14/80 (17.5%) |
| | 9 | 1/200 (0.5%) | 1/80 (1.3%) |
| | 12 | – | 2/80 (2.5%) |

[a] $(GCA)_6{}^{\#}$; [b] $(GCA)_7{}^{\#}$; – not detected.

Furthermore, the haplotypes of two SSR loci of *LaSCL6* were analyzed, and 12 haplotypes occurred, among which $(GCA)_7(CCA)_7$ was predominant (206/280, 73.6%) (Table 2, Figure 1). The occurrence of polymorphism of *LaSCL6* helps to study the diversity of the *L. kaempferi* population.

**Table 2.** Haplotypes of two simple sequence repeats in *LaSCL6* in the population of *Larix kaempferi* (Lamb.) Carr.

| Haplotype | Frequency in Endosperms | Frequency in Seedlings |
|---|---|---|
| $(GCA)_4 (CCA)_7$ | 3/200 (1.5%) | 2/80 (2.5%) |
| $(GCA)_4 (CCA)_8$ | 1/200 (0.5%) | 2/80 (2.5%) |
| $(GCA)_4 (CCA)_9$ | 1/200 (0.5%) | 1/80 (1.3%) |
| $(GCA)_4 (CCA)_{12}$ | – | 1/80 (1.3%) |
| $(GCA)_6 (CCA)_7$ | 2/200 (1.0%) + 15/200 (7.5%)[a] | 4/80 (5.0%) + 4/80 (5.0%) [a] |
| $(GCA)_6 (CCA)_8$ | – | 9/80 (11.3%) |
| $(GCA)_7 (CCA)_6$ | – | 1/80 (1.3%) |
| $(GCA)_7 (CCA)_7$ | 160/200 (80.0%) | 43/80 (53.8%) + 3/80 (3.8%) [b] |
| $(GCA)_7 (CCA)_8$ | 1/200 (0.5%) | 3/80 (3.8%) |
| $(GCA)_7 (CCA)_{12}$ | – | 1/80 (1.3%) |
| $(GCA)_8 (CCA)_7$ | – | 1/80 (1.3%) |
| $(GCA)_9 (CCA)_7$ | 17/200 (8.5%) | 5/80 (6.3%) |

[a] $(GCA)_6{}^{\#} (CCA)_7$; [b] $(GCA)_7{}^{\#} (CCA)_7$; – not detected.

Notably, in the repeat region of GCA, there were two single-nucleotide polymorphisms (SNPs) (G-A and C-T) (Figure 1, Tables 1 and 2). When G is changed to A, the codon CAG changes to CAA, but this is a synonymous SNP without changing the amino-acid. When C is changed to T, the codon CAG changes to the termination codon TAG, resulting in the termination of *LaSCL6* translation and a change in the protein length, and this reveals the regulation of *LaSCL6* expression at the genomic level [22].

**Figure 1.** Information on simple sequence repeats in the genomic DNA sequence of *LaSCL6*. The GCA and CCA repeat regions are underlined. The arrows indicate the loci of single nucleotide polymorphisms (SNPs) ([#] haplotypes with SNPs; * identical residues).

### 3.2. SSRs Affect LaSCL6 Structure and Function

Protein has a defined three-dimensional structure after the folding of the primary structure. The three-dimensional structures of 12 haplotypes of *LaSCL6* were predicted and the results showed that they differed in at least at five places (Figure 2). The GCA and CCA repeat regions in *LaSCL6* result in repeats of the CAG and CCA codons, which code for polyglutamine and proline, respectively, and this might affect the structure and function of *LaSCL6*.

SSRs in the exon affect protein structure and thus lead to a change of protein function [24]. In human genes, tandem repeats of polyglutamine cause incurable neurodegenerative diseases [25,26]. In *Arabidopsis thaliana*, polyglutamine length in *EARLY FLOWERING 3* has dramatic effects on flowering time and circadian clock-related phenotypes [24]. Poly proline affects the binding of profilin and may have consequences for the regulation of actin cytoskeletal dynamics in plant cells [27]. The SSRs in the exon of *LaSCL6* change the primary structure of its protein, and this affects its three-dimensional structure, which might result in the functional diversification of *LaSCL6*; further work is needed to test this.

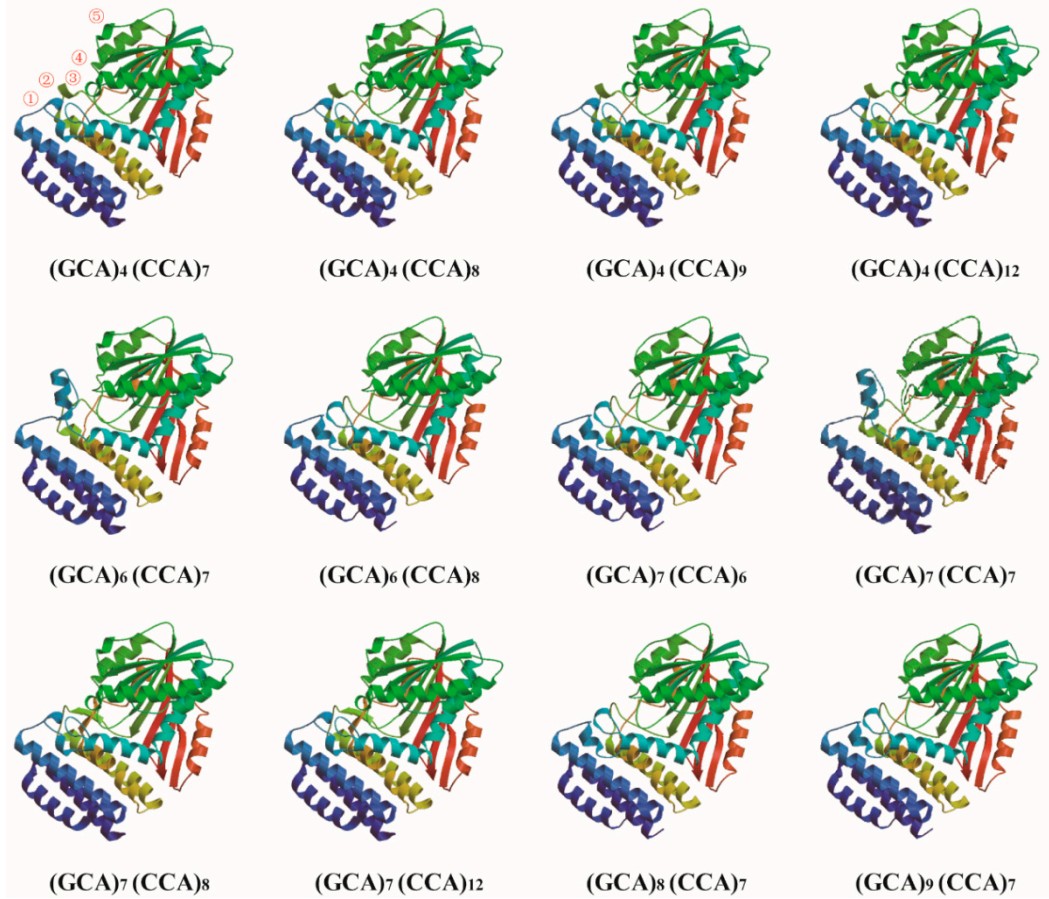

**Figure 2.** Prediction of protein tertiary structures of different haplotypes of LaSCL6. Numbers indicate the potential differences among the structures.

### 3.3. Parent-Offspring Relationships Can Be Analyzed by SSRs in LaSCL6

*L. kaempferi* is monoecious and mainly wind-pollinated [15]. Accurate information about parent-offspring relationships is important for *Larix* breeding programs [10,16]. Three mature trees were used to study the genetic information delivery of *LaSCL6* from parents to offspring based on the haplotypes of the two SSR loci in *LaSCL6*.

Tree 1 was heterozygous and had two haplotypes: $(GCA)_7(CCA)_7$ and $(GCA)_4(CCA)_7$ (Table 3). The haplotypes of endosperms and embryos from 13 seeds were determined. All the haplotypes of 13 endosperms and 92.3% (24/26) of the haplotypes of 13 embryos were found in tree 1, while 7.7% (2/26) were not found in tree 1 (Table 3).

Tree 2 was heterozygous and had two haplotypes: $(GCA)_7(CCA)_7$ and $(GCA)_6(CCA)_7$ (Table 3). The haplotypes of endosperms and embryos from 7 seeds were determined. All the haplotypes of 7 endosperms and 78.6% (11/14) of the haplotypes of 7 embryos were found in tree 2, while 21.4% (3/14) were not found in tree 2 (Table 3).

Tree 3 was homozygous and had one haplotype $(GCA)_7(CCA)_7$ (Table 3). The haplotypes of endosperms and embryos from 9 seeds were determined. All the haplotypes of 9 endosperms and 66.7% (12/18) of the haplotypes of 9 embryos were found in tree 3, while 33.3% (6/18) were not found in tree 3 (Table 3).

**Table 3.** Haplotypes of two simple sequence repeats in *LaSCL6* of mother trees and their seeds.

| Mother Tree | Seed | | |
| --- | --- | --- | --- |
| | Number | Endosperm | Embryo |
| Tree 1 $(GCA)_7 (CCA)_7$ (23/54, 42.6%) $(GCA)_4 (CCA)_7$ (31/54, 57.4%) | 1 | $(GCA)_7 (CCA)_7$ | $(GCA)_7 (CCA)_7$ (10/10) (100.0%) |
| | 2 | $(GCA)_4 (CCA)_7$ | $(GCA)_7 (CCA)_7$ (5/10) (50.0%) $(GCA)_4 (CCA)_7$ (5/10) (50.0%) |
| | 3 | $(GCA)_7 (CCA)_7$ | $(GCA)_7 (CCA)_7$ (5/10) (50.0%) $(GCA)_4 (CCA)_7$ (5/10) (50.0%) |
| | 4 | $(GCA)_4 (CCA)_7$ | $(GCA)_7 (CCA)_7$ (7/10) (70.0%) $(GCA)_4 (CCA)_7$ (3/10) (30.0%) |
| | 5 | $(GCA)_7 (CCA)_7$ | $(GCA)_7 (CCA)_7$ (10/10) (100.0%) |
| | 6 | $(GCA)_4 (CCA)_7$ | $(GCA)_7 (CCA)_7$ (4/10) (40.0%) $(GCA)_4 (CCA)_7$ (6/10) (60.0%) |
| | 7 | $(GCA)_4 (CCA)_7$ | $(GCA)_7 (CCA)_7$ (4/10) (40.0%) $(GCA)_4 (CCA)_7$ (6/10) (60.0%) |
| | 8 | $(GCA)_4 (CCA)_7$ | $(GCA)_7 (CCA)_7$ (6/10) (60.0%) $(GCA)_4 (CCA)_7$ (4/10) (40.0%) |
| | 9 | $(GCA)_7 (CCA)_7$ | **$(GCA)_9 (CCA)_7$ (4/10)(40.0%)** $(GCA)_7 (CCA)_7$ (6/10) (60.0%) |
| | 10 | $(GCA)_7 (CCA)_7$ | **$(GCA)_6 (CCA)_8$ (4/10)(40.0%)** $(GCA)_7 (CCA)_7$ (6/10) (60.0%) |
| | 11 | $(GCA)_7 (CCA)_7$ | $(GCA)_7 (CCA)_7$ (10/10) (100.0%) |
| | 12 | $(GCA)_4 (CCA)_7$ | $(GCA)_7 (CCA)_7$ (5/10) (50.0%) $(GCA)_4 (CCA)_7$ (5/10) (50.0%) |
| | 13 | $(GCA)_4 (CCA)_7$ | $(GCA)_7 (CCA)_7$ (10/10) (100.0%) |
| Tree 2 $(GCA)_7 (CCA)_7$ (58/60, 96.7%) $(GCA)_6 (CCA)_7$ (2/60, 3.3%) | 1 | $(GCA)_7 (CCA)_7$ | $(GCA)_7 (CCA)_7$ (8/10) (80.0%) $(GCA)_6 (CCA)_7$ (2/10) (20.0%) |
| | 2 | $(GCA)_7 (CCA)_7$ | $(GCA)_7 (CCA)_7$ (5/10) (50.0%) **$(GCA)_6 (CCA)_8$ (5/10)(50.0%)** |
| | 3 | $(GCA)_7 (CCA)_7$ | $(GCA)_7 (CCA)_7$ (10/10) (100.0%) |
| | 4 | $(GCA)_7 (CCA)_7$ | $(GCA)_7 (CCA)_7$ (10/10) (100.0%) |
| | 5 | $(GCA)_7 (CCA)_7$ | $(GCA)_7 (CCA)_7$ (9/10) (90.0%) **$(GCA)_8 (CCA)_7$ (1/10)(10.0%)** |
| | 6 | $(GCA)_7 (CCA)_7$ | $(GCA)_7 (CCA)_7$ (8/10) (80.0%) **$(GCA)_7 (CCA)_6$ (2/10)(20.0%)** |
| | 7 | $(GCA)_7 (CCA)_7$ | $(GCA)_7 (CCA)_7$ (9/10) (90.0%) $(GCA)_6 (CCA)_7$ (1/10) (10.0%) |
| Tree 3 $(GCA)_7 (CCA)_7$ (60/60, 100.0%) | 1 | $(GCA)_7 (CCA)_7$ | $(GCA)_7 (CCA)_7$ (9/10) (90.0%) **$(GCA)_7 (CCA)_6$ (1/10)(10.0%)** |
| | 2 | $(GCA)_7 (CCA)_7$ | $(GCA)_7 (CCA)_7$ (9/10) (90.0%) **$(GCA)_7 (CCA)_6$ (1/10)(10.0%)** |
| | 3 | $(GCA)_7 (CCA)_7$ | $(GCA)_7 (CCA)_7$ (8/10) (80.0%) **$(GCA)_4 (CCA)_8$ (2/10)(20.0%)** |
| | 4 | $(GCA)_7 (CCA)_7$ | $(GCA)_7 (CCA)_7$ (9/10) (90.0%) **$(GCA)_6 (CCA)_7$ (1/10)(10.0%)** |
| | 5 | $(GCA)_7 (CCA)_7$ | $(GCA)_7 (CCA)_7$ (5/10) (50.0%) **$(GCA)_6 (CCA)_7$ (5/10)(50.0%)** |
| | 6 | $(GCA)_7 (CCA)_7$ | $(GCA)_7 (CCA)_7$ (10/10) (100.0%) |
| | 7 | $(GCA)_7 (CCA)_7$ | $(GCA)_7 (CCA)_7$ (10/10) (100.0%) |
| | 8 | $(GCA)_7 (CCA)_7$ | $(GCA)_7 (CCA)_7$ (10/10) (100.0%) |
| | 9 | $(GCA)_7 (CCA)_7$ | $(GCA)_7 (CCA)_7$ (9/10) (90.0%) **$(GCA)_6 (CCA)_8$ (1/10)(10.0%)** |

The haplotypes of embryos different from that of their mother trees are shown in bold. The numbers in parentheses refer to the occurrence frequencies of the haplotypes in the sequenced clones.

In the seeds of *L. kaempferi*, the embryo (2n) is diploid and its two haplotypes come from the female and male parents; the endosperm (n) is haploid and its haplotype is from the mother tree and the same as one haplotype of the embryo. Here, we took *LaSCL6* as a case study and found that the haplotype of an endosperm was the same as one haplotype of the mother tree and the embryo. For most seeds, the two haplotypes of an embryo were the same as those of the female parent (Table 3), indicating that the female and male parents had the same haplotype. For some seeds, one haplotype of an embryo differed from those of the female parent (Table 3), indicating that it was from the male parent that had a different genetic component from the female parent. Notably, the same haplotype occurred in female and male parents, suggesting that these two SSR loci in *LaSCL6* make no contribution to self-incompatibility.

### 3.4. The Expressions of LaSCL6 Alleles Have the Same Patterns

Allelic expression imbalance has been studied in several plant species, and sometimes it can result in a phenotypic change [5–7,28]; to determine whether it occurs in *LaSCL6*, the expression patterns of five *LaSCL6* alleles in three heterozygous *L. kaempferi* seedlings were analyzed by qRT-PCR assay with allele-specific primers (Table 4).

**Table 4.** Primers for allele-specific quantitative reverse transcription-polymerase chain reaction.

| Primer Name | Primer Sequence (5′-3′) |
|---|---|
| $(GCA)_4$ | F-AGAAGAGCC**GCAGCAGCAGCA**GAGCCGAGC |
| | R-AGGCGGAGGGAGGTCTTT |
| $(GCA)_6$ | F-CCCTGCACGTTGGTATCCT |
| | R-GCTCGGCTC**TGCTGCTGCTGCTGCTGC**GGCTCTTCT |
| $(GCA)_7$ | F-CCCTGCACGTTGGTATCCT |
| | R-CGGCTC**TGCTGCTGCTGCTGCTGCTGC**GGCTCT |
| $(CCA)_7$ | F-CCTCCG**CCACCACCACCACCACCACCA**TTTGCC |
| | R-CCGAGGCCATGTTAAAGC |
| $(CCA)_8$ | F-CCTCCG**CCACCACCACCACCACCACCACCA**TTTGCC |
| | R-CCGAGGCCATGTTAAAGC |

F, forward primer; R, reverse primer; the repeat regions in primers are shown in bold.

The haplotypes of seedling 1 were $(GCA)_7(CCA)_7$ and $(GCA)_6(CCA)_8$ (Figure 3a). The expression patterns of *LaSCL6* with $(GCA)_7(CCA)_7$ were detected with primer sequences containing the $(GCA)_7$ or $(CCA)_7$ repeat motif, and those of *LaSCL6* with $(GCA)_6(CCA)_8$ were detected with primer sequences containing the $(GCA)_6$ or $(CCA)_8$ repeat motif (Figure 3a, Table 4). For *LaSCL6* with $(GCA)_7(CCA)_7$, the same expression patterns were detected by the $(GCA)_7$ and $(CCA)_7$ primers (Figure 3a, Table 4). For *LaSCL6* with $(GCA)_6(CCA)_8$, the same expression patterns were detected by the $(GCA)_6$ and $(CCA)_8$ primers (Figure 3a, Table 4). For two *LaSCL6* alleles, the same expression patterns were detected by the primers $(GCA)_7$ and $(GCA)_6$, or $(CCA)_7$ and $(CCA)_8$ (Figure 3a, Table 4).

The haplotypes of seedling 2 were $(GCA)_4(CCA)_8$ and $(GCA)_6(CCA)_8$. Almost the same expression patterns for two *LaSCL6* alleles were detected by the primers $(GCA)_4$ and $(GCA)_6$ (Figure 3b, Table 4). The haplotypes of seedling 3 were $(GCA)_4(CCA)_7$ and $(GCA)_7(CCA)_7$, and almost the same expression patterns for two *LaSCL6* alleles were detected by the primers $(GCA)_4$ and $(GCA)_7$ (Figure 3c, Table 4).

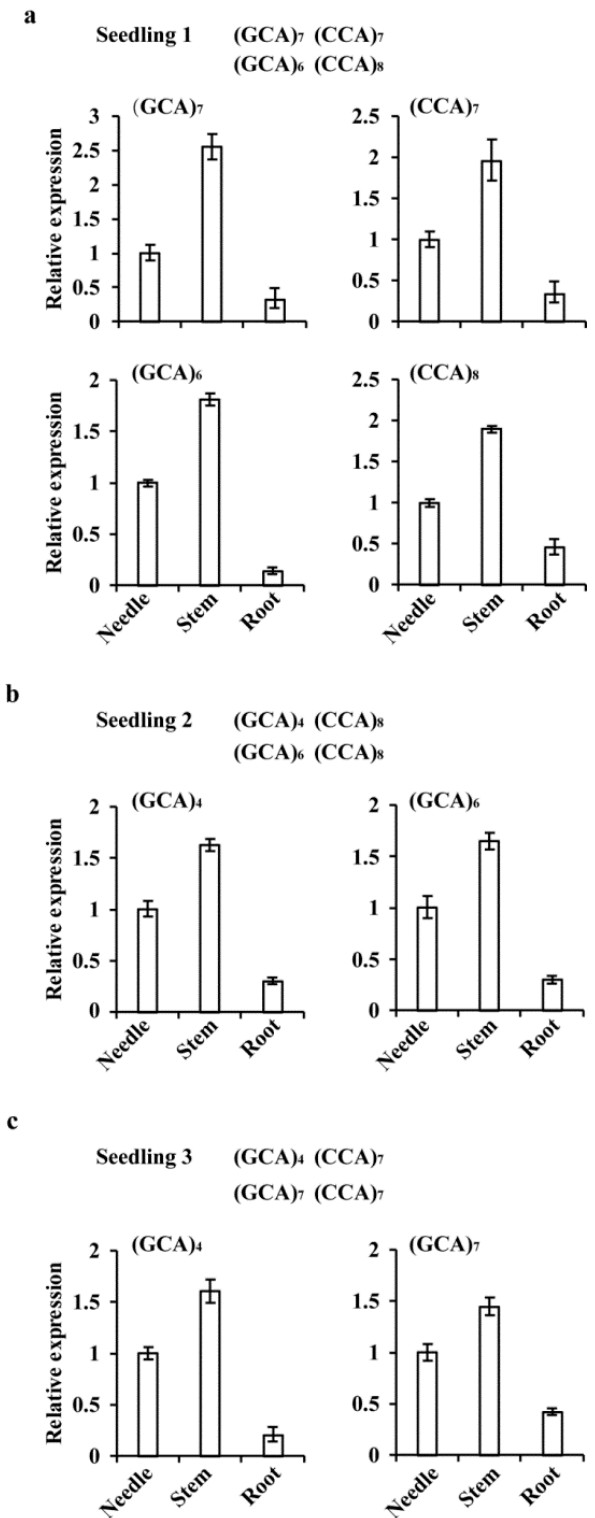

**Figure 3.** The expression patterns of *LaSCL6* alleles. Expression patterns of *LaSCL6* alleles in needle, stem, and root of *Larix kaempferi* (Lamb.) Carr seedling 1 (**a**), seedling 2 (**b**), and seedling 3 (**c**) assayed by qRT-PCR with *LaEF1A1* as the internal control. The qRT-PCR was performed with three biological replicates, and the data are shown as the mean ±SD.

Taken together, all five alleles were strongly expressed in stems, weakly in roots, and showed almost the same expression patterns and balanced expression during the process of vegetative development.

## 4. Conclusions

In summary, the SSRs in *LaSCL6* show high levels of polymorphism and might affect the protein structure and function. Based on an analysis of the haplotypes of *LaSCL6* in mother trees and their offspring, the parent-offspring relationships were determined. These results provide a candidate marker for the *L. kaempferi* genetic studies and breeding.

Altogether, we have obtained new information on *LaSCL6*, especially its regulation by microRNA171 and its genomic structure. For future studies, constructing the LaSCL6 network by mining its interacting proteins and target genes will help to reveal the molecular mechanisms underlying the growth and development of *L. kaempferi*.

**Author Contributions:** Q.-L.Z. carried out the study, analyzed the data, and wrote the manuscript. X.-Y.L. performed the isolation of endosperm and DNA extraction. W.-F.L. designed the study, analyzed the data, and revised the manuscript. L.-W.Q. provided suggestions on the experimental design and analyses. All authors have read and agreed to the published version of the manuscript.

**Funding:** This work was supported by the National Natural Science Foundation of China (31770714), the Basic Research Fund of Research Institute of Forestry (RIF2014-07), and the National Transgenic Major Program (2018ZX08020-003).

**Acknowledgments:** The authors thank IC Bruce (Peking University) for critical reading of the manuscript.

**Conflicts of Interest:** The authors declare no conflict of interest.

**Data Archiving Statement:** The full-length cDNA sequences of *LaSCL6* and *LaSCL6-variant1* have been deposited in GenBank with the accession number JX280920 and MK501379, respectively.

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
