# Peer review of "Identification and Characterization of LaSCL6 Alleles in Larix kaempferi (Lamb.) Carr. Based on Analysis of Simple Sequence Repeats and Allelic Expression"

_forests, doi:10.3390/f11121296_

Round 1

Reviewer 1 Report

The manuscript is of interest for the forest production sector and for the genetics. Please improve the introduction in relation to more others studies in this field.

The Conclusion chapter is completely missing, please reconsider!

Author Response

Thank for your careful reading and good suggestion, we have modified accordingly.

Review 1

Comment: The manuscript is of interest for the forest production sector and for the genetics. Please improve the introduction in relation to more others studies in this field.

Response: We have improved the introduction and added more information. Please check them.

Comment: The Conclusion chapter is completely missing, please reconsider!

Response: We have added Conclusions to the manuscript. Please check them.

Reviewer 2 Report

The study submitted by Zang et al. is focused on the identification and characterization of LaSCL6 alleles in Larix kaempferi using SSR markers and allelic expression. The data obtained are unclear and not well described. It is necessary to emphasize the major outcomes of this research and to suggest possible applications and further researches.

Title

The title does not need modifications.

Abstract

The abstract is written well. Just few suggestions.

Line 16: change “In this study, we analyzed the polymorphisms of these two SSRs in the L. kaempferi population, and found each had five types, among which (GCA)7 and (CCA)7 were predominant.” in “In this study, we analyzed the polymorphisms of these two SSRs in the L. kaempferi population. We found that each SSR had five different polymorphisms, among which (GCA)7 and (CCA)7 were predominant.”

Keywords

Please do not use the same words you have in the title. For example, change “Larix kaempferi” in “Japanese larch”.

Introduction

The introduction is written correctly but too short. Please enrich the text with more information about the material (L. kaempferi) and about the state-of-art of SSR use in this specie. Why do the scientific community need more SSRs in L. kaempferi?

LaSCL6 seems to be targeted from miRNAs, but no information are reported about it in th text.

Line 34: change “special” with “specific”

Line 38: which was the reference sequence that you used to identify LaSCL6?

Line 42: add the subject in “and studied”

Material and methods

2.1 Plant material

Some details are missing. All your samples come from three mature trees? For a better comprehension of your materials description you should avoid “;” and use “.”

Results and Discussion

This section is the one that needs more corrections. The results are unclear and the discussion is very weak.
Which are the implications of your findings? Can you add some suggestions about your future researches?

Figure 1. The resolution is not good, please improve it!

Line 106: You should use some references from plant genetic studies

3.3. Parent-offspring relationships can be analyzed by SSRs in LaSCL6

Why did you used a different number of seeds?

3.4. The expressions of LaSCL6 alleles have the same patterns

Figure 3. The data are not explained! You should add a legend or describe your results in the text. Which are the implications of these expression patterns?

Author Response

Thank for your careful reading and good suggestion, we have modified accordingly.

Review 2

Comment: The study submitted by Zang et al. is focused on the identification and characterization of LaSCL6 alleles in Larix kaempferi using SSR markers and allelic expression. The data obtained are unclear and not well described. It is necessary to emphasize the major outcomes of this research and to suggest possible applications and further researches.

Response: We have added more information on the background in the Introduction, more information on the data in the Results, and more information on the implication in the Conclusions, emphasizing the background and the major outcomes of this research and suggesting possible applications and further researches. Please check them.

Abstract

Comment: Line 16: change “In this study, we analyzed the polymorphisms of these two SSRs in the L. kaempferi population, and found each had five types, among which (GCA)7 and (CCA)7 were predominant.” in “In this study, we analyzed the polymorphisms of these two SSRs in the L. kaempferi population. We found that each SSR had five different polymorphisms, among which (GCA)7 and (CCA)7 were predominant.”

Response: We have changed these sentences. Please check them.

Keywords

Comment: Please do not use the same words you have in the title. For example, change “Larix kaempferi” in “Japanese larch”.

Response: We have changed the Keywords. Please check them.

Introduction

Comment: The introduction is written correctly but too short. Please enrich the text with more information about the material (L. kaempferi) and about the state-of-art of SSR use in this specie. Why do the scientific community need more SSRs in L. kaempferi?

Response: We have added more information about the SSRs and its application in L. kaempferi in the Introduction. Please check them.

Comment: LaSCL6 seems to be targeted from miRNAs, but no information is reported about it in the text.

Response: We have added the information about its regulation by microRNA171 in the Introduction. Please check them.

Comment: Line 34: change “special” with “specific”

Response: We have changed this word. 

Comment: Line 38: which was the reference sequence that you used to identify LaSCL6?

Response: In the previous study [Tree Genetics & Genomes, 2014. 10(1): p. 223-229], according to the cDNA sequences of SCL6 homologs from P. taeda (TC67393) and P. glauca (BT102743.1), we cloned the full-length cDNA sequence of L. kaempferi homolog, LaSCL6 (JX280920, 2295 bp).

In another study [Tree Genetics & Genomes, 2019. 15:57], based on the transcriptome data, a longer transcript (3073 bp), named LaSCL6-variant1 (MK501379), was cloned. Then these two transcripts were further proved to be alternative variants transcribed from one DNA sequence (3073 bp).

Comment: Line 42: add the subject in “and studied”

Response: We have added the subject here.

Material and methods

2.1 Plant material

Comment: Some details are missing. All your samples come from three mature trees? For a better comprehension of your materials description you should avoid “;” and use “.”

Response: Only the samples used to study the genetic information delivery from parents to offspring were collected from three mature trees, including the needles and seeds. We did not change the “;” to “.”. When used “.”, the information of materials will be mixed.

Results and Discussion

Comment: This section is the one that needs more corrections. The results are unclear and the discussion is very weak.

Response: We have modified the Results and Discussion, and added the Conclusions. Please check them.

Comment: Which are the implications of your findings? Can you add some suggestions about your future researches?

Response: We have added the implications of our findings and suggestions about future researches in the Conclusions. Please check them.

Comment: Figure 1. The resolution is not good, please improve it!

Response: We have modified the Figure 1 with high resolution (1200 dpi). Please check it.

Comment: Line 106: You should use some references from plant genetic studies

Response: As far as we know, the similar findings in Arabidopsis thaliana and maize are reported, and we have cited them (reference 24 and 27).

3.3. Parent-offspring relationships can be analyzed by SSRs in LaSCL6

Comment: Why did you use a different number of seeds?

Response: We selected the needles of these three trees in July to detect the SSRs in LaSCL6. When we harvested their seeds in August, we found that the seed yield and quality were poor, resulting in fewer seeds that can be used.

3.4. The expressions of LaSCL6 alleles have the same patterns

Comment: Figure 3. The data are not explained! You should add a legend or describe your results in the text. Which are the implications of these expression patterns?

Response: We have added a legend for Figure 3. and these results are described from line 150-178. Here we only want to show whether the expression of LaSCL6 alleles have the balanced patterns, as to its function in the formation of needle, stem, and root, we do not study.